# Mechanisms of Social Attachment Between Children and Pet Dogs

**DOI:** 10.3390/ani14203036

**Published:** 2024-10-20

**Authors:** Olivia T. Reilly, Leah H. Somerville, Erin E. Hecht

**Affiliations:** 1Department of Human Evolutionary Biology, Harvard University, Cambridge, MA 02138, USA; erin_hecht@fas.harvard.edu; 2Department of Psychology, Harvard University, Cambridge, MA 02138, USA; somerville@fas.harvard.edu; 3Center for Brain Science, Harvard University, Cambridge, MA 02138, USA

**Keywords:** human-animal interaction, social attachment, cortisol, oxytocin, children, dogs

## Abstract

**Simple Summary:**

There is evidence to suggest that the strength of the social attachment that forms between a human and their pet dog is important for maximizing the therapeutic outcomes of pet dog ownership, more so than the presence of a dog alone. Here, we review the literature to determine whether this evidence is supported specifically in children with pet dogs. We discuss the benefits to child health and well-being that are associated with pet dog ownership, the neural and endocrinological mechanisms that may support these intra-species attachments, and the importance of taking a dyadic approach to the study of this topic in the future.

**Abstract:**

An increasing body of evidence indicates that owning a pet dog is associated with improvements in child health and well-being. Importantly, the degree of the social bond between child and dog may mediate the beneficial outcomes of dog ownership. The formation of social bonds is an intrinsically dyadic, interactive process where each interactor’s behavior influences the other’s behavior. For this reason, it is critical to evaluate the biological mechanisms of attachment in both children and their pet dogs as a socially bonded pair. Here, we review the physical, mental, and emotional outcomes that are associated with pet dog ownership or interaction in children. We then discuss the evidence that suggests that the strength of a social bond between a child and their pet dog matters for maximizing the beneficial outcomes associated with pet dog ownership, such as possible stress-buffering effects. We review the existing literature on the neural and endocrinological mechanisms of social attachment for inter-species social bonds that form between human children and dogs, situating this emerging knowledge within the context of the mechanisms of intra-species bonds in mammals. Finally, we highlight the remaining open questions and point toward directions for future research.

## 1. Introduction

Pets play an important role in child health and well-being [1]; dogs specifically enhance this well-being more so than other animals [2]. The experience of stress in pre-adolescence can have life-altering effects into adulthood, possibly differentially affecting males and females [3]. Children with pet dogs experience physical, social, and mental health benefits [4,5,6,7,8,9,10]. However, minimal work has examined the factors that lead to strong attachment bonds compared to weaker ones between pet dogs and children, despite evidence that suggests that the degree of the social bond between child and their pet dog may matter for therapeutic outcomes more so than having a pet dog in the first place [11,12]. Little attention has been devoted to the study of both sides of the child–dog dyad with regard to beneficial outcomes: the canine behavioral and endocrinological literature has been largely disconnected from the neurobiological literature, which has primarily focused on other non-canine species, such as rodent and nonhuman primate species. Bridging this gap will provide insight into the probable neural mechanisms underlying the dog/human interaction, as well as highlight missing information and targets for future research.

This review has three overarching goals. The first is to summarize the current literature on the benefits of social attachment formation with non-clinical populations of children and their pet dogs. Many studies focus on the therapeutic benefits of dog ownership on children with a clinical diagnosis or in a medical setting [13,14,15,16,17,18,19,20], but far less is known about how social attachment to a pet dog influences non-clinical populations of children. There is a critical need for the evaluation of beneficial outcomes of pet dog ownership on non-clinical populations of children to build a foundation that informs the clinical child population literature. Further, there may be benefits in non-clinical populations of children that extend to clinical populations, as variation exists along a continuum in behavior and mental function [21,22].

The second goal of this review is to highlight the need for studies that do not focus solely on the beneficial outcomes of pet ownership on children but also on the impact of social attachment on the pet dog. Some studies have investigated well-being outcomes in dogs (see [23] for review), but many studies to date lack information about both sides of the dyad. It is clear that there are increased therapeutic outcomes associated with dog ownership in children, particularly when the attachment of the child to the dog is strong [12,24]. However, it is unclear whether this perceived attachment is reciprocated by the dog, what the mechanisms of social attachment development are in dogs, and to what extent the quality of the social bond influences therapeutic outcomes for children. An attachment bond necessarily involves a dyad, and an important possibility is that potential outcomes for the child cannot be fully understood without also examining outcomes for the dog.

The final goal of this review is to obtain a better understanding of the mechanisms that underlie social attachment formation and its beneficial effects on mental health in children and pet dogs in order to maximize the therapeutic capacities of dog ownership for children, extending to clinical populations of children in the future. One proposed mechanism for the benefits of social attachment to pet dogs is that the social bond buffers the stress response [25]. This social buffering hypothesis has been well studied in adults but less so in children. This review highlights potential future avenues for the study of mechanisms that underlie the child–dog social attachment literature.

## 2. Behavioral Perspective

### 2.1. Physical Activity

Dog ownership yields many benefits for children in non-clinical settings. For instance, children show increases in physical activity [4,26,27] with pet dog ownership. A group of 727 children in Australia aged 10–12 years that walked their dog were also more likely to play outside and were more independent than children who did not walk their dog [27]. In 14 year old adolescents, a positive association was found between dog ownership and physical activity level [26].

### 2.2. Learning in the Classroom

Several studies suggest that children show improvements in social and learning outcomes from being in the presence of dogs. In a dog-assisted reading program, children ages 6–8 showed improvements in reading ability when reading out loud to a dog in the classroom, when spending 15–20 min training the dog, and when a dog was passively present in the classroom [5]. In preschool-aged children (3–5 years old), the presence of a dog in the classroom reduced the number of instructional prompts required for children for the completion of cognitive tasks [6].

### 2.3. Learning and Development at Home

While the presence of a dog can yield beneficial outcomes for children in the classroom, living with a pet dog in the home is thought to provide even greater improvements in child well-being. Owning a pet dog with an infant child in the household was associated with a decrease in the risk of the infant experiencing developmental delays [28]. Children may be more motivated when in the presence of a pet dog due to the belief that the dog gives them full attention [29]. Consequently, this can lead to an increase in a child’s feeling of importance and in their desire to learn more [29].

### 2.4. Mental Health

Dog ownership and companionship are associated with mental health improvements for children. Feelings of loneliness and isolation from others are reduced [8,9,10] in children and adolescents with a canine companion, particularly for homeless youth [9,10]. Similarly, reductions in anxiety [30] or reductions in the probability of developing childhood anxiety [31] are reported with dog ownership. Stress is a major health concern in children, particularly today, as children are faced with social challenges brought on by the pandemic, such as missed opportunities for face-to-face social interaction during a period of development when social and emotional behaviors and skills are still developing. These challenges include transitioning to online learning, social isolation, and general reductions in face-to-face social interactions. Children in the presence of a dog show reductions in stress [14,32,33] assessed via cortisol level, although the neural mechanisms of this relationship are not fully understood.

### 2.5. Relationship Quality Matters

In human relationships, social support does not universally equate to improved health outcomes. Instead, it is the quality of the social support that makes the ultimate difference [34,35,36,37,38]. Not all relationships are created equally: New mothers who have a high-quality relationship with their spouse are more responsive to their infants [36]. Couples in “good-quality” relationships but not “poor-quality” relationships confer mental health benefits over those who are single [35]. Positive relationship quality is associated with lower cardiac output in women [34], and even the perception of strong social support is associated with reductions in loneliness [37,38]. This pattern is not unique to human relationships and appears to extend to human-dog relationships as well.

Evidence suggests that dog ownership alone may not be sufficient to yield beneficial physical health outcomes, but instead, it is the quality of the relationship that matters. For instance, Poresky and Hendrix (1990) [24] conducted a survey study to investigate the role of pet attachment level on developmental measures such as empathy, cooperation, and social competency in children ages 3–6 years old [24]. They found significant correlations between children’s social bond scores and empathy scores, as well as an inverse relationship between social bond scores and uncooperative scores. This suggests that the attachment with the pet, more so than the passive presence of the pet alone, is related to more optimal developmental outcomes. However, this study could not draw any causal conclusions due to the correlative nature of the data. Another study found that 4–10-year-old children with higher levels of social attachment to their dog, as measured by the Companion Animal Bonding Scale [39], were associated with an increase in time spent being active [4]. Additionally, 9–10-year-old children with higher reported social attachment to their dog walked their dog more frequently, which suggests that strong social bonds to pet dogs could increase physical activity beyond dog ownership [40].

Stronger attachment bonds between children and their pet dogs also lead to an increase in therapeutic health outcomes, particularly for children younger than 15 years old [2]. Children with stronger attachment to their dogs scored higher on perceived health and happiness [2], likely due to the reported emotional reciprocity that children experience from their dogs compared to other species of pets. Stronger social attachment to pet dogs is also associated with greater confidence levels in 8–12-year-old children [12]. Increasingly, dogs are being utilized for therapeutic purposes with children [41,42,43,44], and social attachment formation is likely relevant for therapeutic effects to emerge in a maximally effective manner. However, little work has examined the factors that lead to strong attachment bonds compared to weaker ones between pet dogs and children, despite evidence that suggests that the degree of the social bond between child and their pet dog may matter for therapeutic outcomes more so than having a pet dog in the first place [11,12]. It also remains unclear as to whether the mechanisms of the social attachment formation between a child and their dog resemble other mammalian attachment bonds.

## 3. Biological Perspective

### 3.1. Dog-Human Bonds Rely on the Neural and Hormonal Mechanisms That Support Parent-Child Bonds

Maternal bonds set the foundation for future social bonds that an individual may form throughout development, and this is true across the mammalian realm [45,46,47,48]. There are similarities in the physiology and neurobiology of mother-infant bonds, pair bonds, and affiliative attachment bonds [49,50,51]. The physiological correlates of social attachment formation are well studied in mammals and centrally involve the neuropeptide hormone oxytocin. The oxytocin system interacts with brain areas involved in emotion and attachment, including the amygdala, bilateral insula, rostral dorsal cingulate gyrus [52], anterior/midcingulate cortex, and reward processing areas, including the striatum [53,54]. Many of these brain areas are activated in human mothers in response to viewing their human children and family dog’s faces [55], suggesting that the oxytocinergic system is involved in social attachment formation on a neural level in both human and human-dog relationships in adults. In fact, the human-dog social bond greatly resembles the mother-infant social bond [56,57].

The caregiver-infant bond is critical for the development of secure social attachments that infants will form later in life [58,59]. This bond is also heavily dictated by oxytocinergic mechanisms [59]. This is not all that surprising, as oxytocin is centrally involved in the formation and maintenance of mother-infant bonds in mammals generally [50,60], and more specifically, is the proposed hormonal mechanism by which humans and dogs coevolved to form strong social bonds [57,61,62]. Additionally, oxytocin can be reliably and non-invasively measured in salivary forms in dogs [63,64,65] and humans [65,66,67,68], which makes it an ideal hormone for the study of social bond formation.

In dogs, exogenous oxytocin promotes positive social behavior and affiliation towards other dogs and towards human owners [62]. Exchange of these positive social behaviors between dogs creates an increase in endogenous oxytocin levels, showing that oxytocin is involved in the development of social bonds between dogs [62] as well as between humans and dogs, who show elevated oxytocin levels after a bout of prolonged eye contact [61,69].

### 3.2. Oxytocin Modulates Neural and Hormonal Responses to Stress

Of particular importance to children, oxytocin is also thought to have stress-buffering effects [70,71,72,73,74]. Oxytocin inhibits HPA axis activity [75,76,77], which may explain how strong social bonds and attachment formation act as a buffer against the aversive effects of stress in childhood. On a neural level, oxytocin is largely produced by oxytocin neurons in the paraventricular nucleus (PVN) and supraoptic nucleus (SON) of the hypothalamus [50]. The projections of these cells span the whole brain, including the brainstem and limbic areas, which contain glucocorticoid receptors involved in the stress response [78]. Oxytocin receptors are upregulated by glucocorticoids after extended bouts of stress in rodents [78], which explains how oxytocin may regulate the HPA axis and result in stress-buffering effects.

### 3.3. Neural Mechanisms of Attachment Are Likely Similar Across Species

The subcortical structures of the brain that govern sociality are thought to be broadly conserved across mammalian species [79]. As such, dog brains likely support social bonding and attachment in a similar way to primates and rodents. For instance, humans and nonhuman primate species share similar neural mechanisms involved in visual face perception [80,81], and recent studies suggest that there may be similarities that extend to the canine brain as well [82,83,84,85]. Specific to the influence of human attachment on neural activity in dogs, a recent study focused on a population of working dogs, which showed caudate activation in response to viewing images and videos of their familiar human compared to unfamiliar ones [86]. These findings aligned with the results of previous studies that have found associations between face perception and temporal cortex activation [84], as well as caudate activation in dogs [87]. When observing positive social interactions between a caregiver and an unfamiliar dog or a stranger and an unfamiliar dog, pet dogs show activation in the hypothalamus, which suggests that the dogs may have perceived the positive interaction of their caregiver with another dog as a rival to their social attachment [88]. When dogs view their caregiver, they show increased activation in the hippocampal and rostral cingulate gyrus compared to when viewing a familiar non-caregiver or a stranger [52]. These brain areas are associated with mother-infant attachment behaviors in mammals [89]. Other areas are associated with attachment as well. In human mothers, the right posterior superior temporal sulcus (STS) was sensitive to videos of those mothers interacting with their own children synchronously using magnetoencephalography (MEG) methods [90,91]. In fMRI studies, other regions, including the orbitofrontal cortex, right fusiform gyrus, and right insula, showed increased activation in “own” compared to “unfamiliar” interaction conditions [92,93,94,95]. The same pattern has been found in children viewing videos of interactions with their parents compared to unfamiliar individuals using MEG [96]. Children showed increased oscillations in brain regions that respond to attachment stimuli in adults, namely, the right temporal and insular cortex [96], areas that are associated with various social functions. Despite these parallels, there are still areas in need of research to determine the mechanisms of inter-species social bonds and how social bond strength modulates the beneficial outcomes of human-dog interaction.

### 3.4. Neural and Hormonal Mechanisms of Stress and Stress-Buffering

Functional connectivity of stress-related brain circuitry is vulnerable to the effects of stress in humans. Cortical regions that are associated with stress, anxiety, and elevated glucocorticoid levels in humans include the prefrontal cortex, hippocampus, and amygdala [97]. Stress disrupts glucocorticoid output [97], resulting in direct effects on stress-related brain regions. Children are a particularly vulnerable population to the effects of stress: adolescent brains are even more susceptible to the effects of heightened glucocorticoid levels than adults and infants [3]. Because children are young and still developing, they must often rely on external support to aid in appropriate social regulation of the stress response [25].

Surprisingly, little is known about the mechanisms behind the social attachment that forms between children and their pet dogs, despite the known therapeutic outcomes on child development that occur. Of the studies that have investigated the hormonal mechanisms involved in the child–dog attachment process, cortisol and oxytocin have emerged as hormones of interest, perhaps due to the role that they play in social affiliation and the stress-buffering response. Cortisol is the primary output of the hypothalamic-pituitary-adrenal (HPA) axis [98,99], and cortisol secretion varies in response to environmental or psychological stress [100]. The HPA axis increases activity in response to these stressors. Thus, cortisol is often used as a quantifiable index of stress [70]. Cortisol levels have been successfully assessed via salivary measures in humans [25,101,102] and dogs [101,103], which makes this an ideal hormone to evaluate stress in both species.

### 3.5. Social Buffering and Stress Reduction

It is generally accepted that exposure to stressors in childhood and adolescence may play a significant role in the formation of psychological morbidities later in life [104], emphasizing the need for a more comprehensive understanding of preventative measures that can be taken to combat the aversive impact of stressors on child health and development. One such measure manifests in the form of social support. The social buffer hypothesis states that social support can act as a protective factor against the aversive effects of stress [70,105]. Several studies have investigated social buffering effects in humans, many in the context of parent-child attachment relationships [106]. However, few studies have investigated this relationship specifically in children and their pet dogs, and even fewer have investigated the physiological mechanisms in dogs as well as children. In children ages 7–12 years, children perceived less stress in the presence of their pet dog compared to in a condition with a parent or being alone [25], based on self-reports, suggesting that children experience the benefits of social buffering in their relationship with their dog. However, variability existed within children’s cortisol responses. During a stress-inducing condition, lower cortisol levels in children were associated with greater child-initiated petting of their dog and lower levels of proximity-seeking behavior from the dog, which provides evidence of the social and emotional support that the presence of a pet dog can provide in a stressful scenario [25]. One recent study investigated the physiological response of 8–10-year-old children when interacting with their pet dog or an unfamiliar dog in a naturalistic setting. The study found that children showed a reduction in salivary cortisol levels after interacting with dogs compared to baseline [107], as did the dogs that they interacted with. Interestingly, this effect was most pronounced in children when they interacted with the unfamiliar therapy dog compared to their pet dog and was most pronounced in pet dogs compared to the unfamiliar dog. The most evident reductions in cortisol were associated with stronger social attachments, determined by using surveys that evaluated the human-animal bond [107]. This shows that child–dog interactions can reduce HPA activity in both dogs and children, particularly when attachment levels are more pronounced. Future avenues to explore include investigating socially attached children and dogs using a physiological measure of attachment, such as salivary oxytocin level.

### 3.6. Dyadic Approach to the Study of Social Attachment

No studies to date have investigated mechanistic changes in the development of social bond formation in children and in their pet dogs simultaneously. Children form social attachments with their pet dogs [108], and the level of this attachment can vary between child–dog dyads. There is evidence that the level of social bond/attachment that has formed between an individual and their dog can dictate the degree to which the individual will experience the buffering effects of that companion against stressors, though this literature primarily focuses on outcomes from just the child side of the child–dog dyad [24,25,109]. Horn and colleagues (2013) evaluated whether familiarity with a human social interactor or the specific relationship to a human social interactor was more important in predicting dog attention toward humans. The relationship between the human interactors and dogs was based on owner reports about caretaking behaviors toward the dog, such as feeding, walking, and playing with the dog. Each of the two human caretakers was sorted into one of the two following groups: “responsibility shared” or “responsibility not shared” based on their report responses. Each dog experienced three trials. In one trial, the first familiar human was manipulating three different targets. In the second trial, the second familiar human manipulated the three different targets, and in the third trial, the unfamiliar experimenter manipulated the three targets. After 30 s of observing a human manipulate the three different targets as described above, the dog was released and allowed to approach any of the targets. Dogs looked longer at their human caretakers with shared responsibility compared to the unfamiliar experimenter. Additionally, when caretakers did not share equal care responsibility, dogs looked longer at the primary caretaker compared to the other familiar caretaker or the unfamiliar experimenter [109]. This demonstrates that the specific relationship that a dog has with a human caretaker is more important than social familiarity in explaining dog attention towards humans. What remains unanswered from this study is how this result might differ with children instead of human adults, as the minimum age to participate was 14 years old.

Social bond formation is an intrinsically dyadic, interactive process where each interactor’s behavior (and underlying neural and endocrine responses) influences the other’s; therefore, it cannot be mechanistically understood by examining only one half of the dyad. For this reason, it is critical to also evaluate the biological mechanisms of attachment in the dog as well as the child (Figure 1). It is possible that a failure of the dog to form an attachment toward the child could inhibit bond formation in the child toward the dog, thus preventing the stress-buffering effects of this bond from occurring. Taken together, the research to date suggests that variations in dog hormonal, neural, and behavioral responses toward the child are necessary for understanding variations in the strength and consequences of bond formation. Understanding the development of the underlying mechanisms of social bond formation in children and dogs could lead to the identification of biomarkers that may be critical for efforts to maximize the beneficial impacts of child–dog bonds in both typical and clinical populations. Previous studies have investigated behavioral changes associated with social bond formation between adults and service dogs but not the underlying endocrine or neural mechanisms [110]. Future studies should focus on child–dog dyads and the underlying mechanisms that support strong social bonds between the two.

## 4. Directions for Future Study

There are still several unanswered questions about the child–dog attachment literature that warrants further investigation in order to be properly addressed (Table 1). Although there is evidence of variation in attachment quality across human-dog dyads [24,25,109], it remains unclear to what degree this variation is reflected in neuroendocrine correlates and how this might influence stress-buffering capacities in child–dog dyads. Additionally, it is not clear to what degree these neuroendocrine biomarkers of attachment are correlated across both individuals in the dyad. Although pet dogs can provide health benefits to children who live with them, less is understood about how dogs might benefit from their child companions. One study did account for the well-being of pet dogs that live with children, though it did not measure outcomes from both perspectives of the dyad at once [111]. This study compared the quality of life for dogs living with neuro-typically and neuro-atypically developing children between the ages of 4–10 years. Based on parent responses to questionnaires, living in a household with neurotypical and neuro-atypical children had many positive outcomes for pet dogs, including the implementation of routine and exercise for the dog and child through engagement in high-energy activities [111]. There were some negative outcomes in both groups as well, such as an increase in exposure to stressors (i.e., tantrums). Ultimately, more work in this area is needed in order to fully address this question.

Another survey study reported that whether the child contributes to taking care of the dog is a predictor of behavioral response from the dog on a pointing gesture task, and performance on the gesture task was associated with the child’s social attachment level to the dog [112]. This suggests that the well-being outcomes can benefit both members of the child–dog dyad. Some promising evidence of a bidirectional attachment has come from the literature on dogs and human adults [113]. Adult attachment style (secure, anxious, or avoidant) was evaluated via the Adult Attachment Questionnaire, followed by observation of the adult-dog dyad across conditions meant to induce challenging situations, such as a stranger approach or a person in costume approach, and presentation of non-social visual and auditory stressors. Subsequently, dog behavior was correlated with human attachment style in several conditions [113], which suggests that owner attachment toward their dog may influence the dog’s behavior toward its owner. However, this work was correlational and could not measure causation, and the sample of dogs was constrained to just one breed. Nonetheless, this work has yielded insights into the dyadic attachment dynamic.

Although we are beginning to see behavioral evidence of positive dyadic outcomes from the social bonds formed between children and their pet dogs, as discussed above, future studies should focus their investigation on elucidating the higher-order social brain regions that might be influenced by this social bond formation in the canine brain. There is a large literature that discusses the biological basis of the oxytocinergic system in the maternal brain in the context of mother-infant bonding [114,115], but no research to date has investigated this topic in the canine brain. Importantly, the dynamic time course of social attachment formation in the brains of both dogs and humans is unknown and should be taken into account in future studies.

## 5. Conclusions

Investigation into the underlying attachment mechanisms of both sides of the child–dog dyad is an area in need of further study. Gaining a better scientific understanding of this topic will provide clarification on the role of social attachment strength in promoting positive health outcomes in children as well as in pet dogs through a dyadic approach. Already, we know that there are clear physical, mental, and emotional benefits for children who form social attachments with their pet dogs. It is likely that the physiological mechanisms of these attachments involve the hormones cortisol and oxytocin and may also resemble the physiology of other types of attachment bonds, such as parent-infant bonds. Future work should also focus on uncovering the neural networks involved in these attachment bonds and whether they resemble other forms of attachment bonds, such as those that occur between adult human handlers and dogs or other bonds that occur more broadly across the mammalian realm. Ultimately, canine behavioral and brain science is a rapidly evolving field with the potential for important contributions to research on cognition, as well as other topics such as aging and emotional well-being.

## Figures and Tables

**Figure 1 animals-14-03036-f001:**
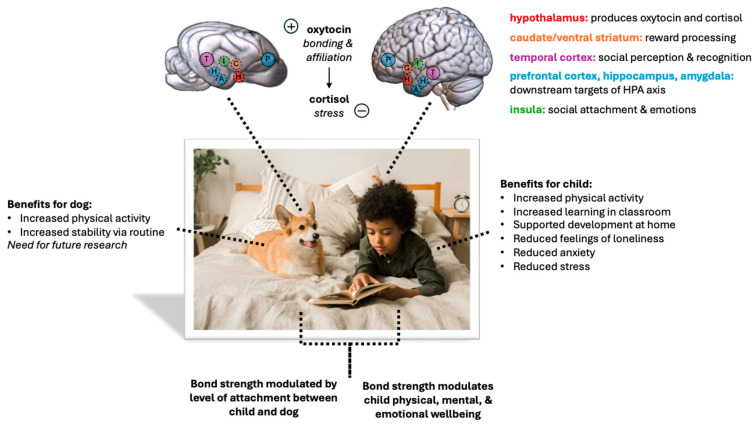
Visual overview of the benefits and mechanisms that underlie child–dog social attachment. Regions of the brain located on the lateral cortical surface are indicated with larger, solid-outlined circles, while regions located beneath the lateral cortical surface are indicated with smaller, dashed circles.

**Table 1 animals-14-03036-t001:** Directions for future research.

Is variation in social attachment strength reflected in neuroendocrine correlates in child–dog dyads?
2. Within a child–dog dyad, what are the outcomes for the dog specifically?
3. Is social attachment of child toward dog always reciprocated by the dog?
4. Is the stress-buffering effect of dogs consistent across social and environmental contexts, including inside and outside the laboratory environment?
5. Does variation in attachment strength/style impact stress-buffering capacities of the social attachment?
6. What higher-order social brain regions are influenced by social attachment formation/oxytocinergic networks in the canine brain?
7. What is the time course of attachment formation in the brain of both dogs and humans?
8. Do mechanisms of social attachment between children and dogs resemble other mammalian attachment bonds?

## Data Availability

No new data were created or analyzed in this study. Data sharing is not applicable to this article.

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
