# Peer review of "Mechanisms of Social Attachment Between Children and Pet Dogs"

_animals, 2024, doi:10.3390/ani14203036_

Round 1
Reviewer 1 Report
Comments and Suggestions for Authors
This narrative review focusses on the attachment relationship between children and their companion dogs, for which considerably less research exists than for the adult human-dog relationship. The authors stated three goals, all of which have been achieved: 1) to review the existing literature on social attachment formation between non-clinical populations of children and their dogs; 2) to promote further research on the dog side of the child-dog dyad, which examines impacts on dog wellbeing; and 3) to better understand and encourage further research on the mechanisms underlying the child-dog relationship.
Indeed, this well-organized review cites over 100 references, all of which are quite relevant to the topic. Both the behavioural and biological perspectives related to child-dog relationships are reviewed, although more emphasis is placed on the biological perspective, which is fair, given that this is the area in which there exists more relevant research, and also that one of the stated goals of the review is to champion further work on the mechanisms of any therapeutic effects of the relationship for both child and dog. One figure depicts both the benefits of and mechanisms that underlie the child-dog bond- although the figure caption suggests that the figure deals only with mechanism, so the authors might wish to update that. The authors then suggest directions for future research, supported by Table 1, which contains eight questions that remain to be addressed. Finally, the authors conclude by reiterating the importance of understanding both sides of the dyadic child-dog attachment relationship, and further suggest other broad questions to be answered in the context of inter-specific social attachment.
I believe this review will encourage further research on the mechanistic basis of child-dog social attachment, and is therefore potentially quite significant in its impact on the field. Given the excellence of the writing, I have only two suggestions regarding revisions:
- Lines 54, 331, perhaps other places?- replace “underly” with “underlie”
- Lines 111, 130- use of “moreso” vs. “more so”- there is controversy about which is correct, but I would simply recommend consistency
I congratulate the authors on this excellent review and look forward to the research that it will likely stimulate!
Author Response
Reviewer 1
This narrative review focusses on the attachment relationship between children and their companion dogs, for which considerably less research exists than for the adult human-dog relationship. The authors stated three goals, all of which have been achieved: 1) to review the existing literature on social attachment formation between non-clinical populations of children and their dogs; 2) to promote further research on the dog side of the child-dog dyad, which examines impacts on dog wellbeing; and 3) to better understand and encourage further research on the mechanisms underlying the child-dog relationship.
Indeed, this well-organized review cites over 100 references, all of which are quite relevant to the topic. Both the behavioural and biological perspectives related to child-dog relationships are reviewed, although more emphasis is placed on the biological perspective, which is fair, given that this is the area in which there exists more relevant research, and also that one of the stated goals of the review is to champion further work on the mechanisms of any therapeutic effects of the relationship for both child and dog. One figure depicts both the benefits of and mechanisms that underlie the child-dog bond- although the figure caption suggests that the figure deals only with mechanism, so the authors might wish to update that.
We appreciate these kind comments! We thank the reviewer for the suggestion about the caption in Figure 1. We have updated this to reflect both the benefits and mechanisms of the child-dog bond.
The authors then suggest directions for future research, supported by Table 1, which contains eight questions that remain to be addressed. Finally, the authors conclude by reiterating the importance of understanding both sides of the dyadic child-dog attachment relationship, and further suggest other broad questions to be answered in the context of inter-specific social attachment.
I believe this review will encourage further research on the mechanistic basis of child-dog social attachment, and is therefore potentially quite significant in its impact on the field. Given the excellence of the writing, I have only two suggestions regarding revisions:
- Lines 54, 331, perhaps other places?- replace “underly” with “underlie”
Thank you for catching this! We have made this replacement throughout the manuscript.
- Lines 111, 130- use of “moreso” vs. “more so”- there is controversy about which is correct, but I would simply recommend consistency
We agree that consistency is key, thank you for pointing this out. We have made sure that we are consistent throughout the manuscript.
I congratulate the authors on this excellent review and look forward to the research that it will likely stimulate!
Thank you!
Reviewer 2 Report
Comments and Suggestions for Authors
-Not sure if this is the final format but I don’t see where the keywords are listed.
-Citations should be in brackets [1] [2] [3] [4-10]
-References are in a different font. Make sure to go through and ensure this is the correct format for the article you are submitting.
-I see that you have two articles that are focused on human-animal interactions. Wondering if there was a reason that human-animal bond research was not discussed or referenced in the review.
-Behavioral Perspective… You can change each category into a subsection such as 2.1 physical activity 2.2 learning in the classroom or you can create an overarching them for groups of topics for example “2.1 skills” and then discussing physical activity, learning in the classroom, and learning and development at home
-Same feedback for Biological Perspective that I had for Behavioral Perspective
-Consider changing main subjects to “Behavioral Perspective” and “Biological Perspective”
-Figure 1: make sure to create an image that can be moved with text and is in line with the rest of the body
-Review requirements for attaching a table and edit Table 1
-Reword conclusion: give brief summary of what was discussed. Change questions into topics/statement of the things that need to be addressed in future
Author Response
Reviewer 2
-Not sure if this is the final format but I don’t see where the keywords are listed.
We thank the reviewer for bringing this to our attention. We have added keywords to the manuscript.
-Citations should be in brackets [1] [2] [3] [4-10]
We have updated the citation style throughout the manuscript.
-References are in a different font. Make sure to go through and ensure this is the correct format for the article you are submitting.
We have fixed the font, and corrected the formatting errors.
-I see that you have two articles that are focused on human-animal interactions. Wondering if there was a reason that human-animal bond research was not discussed or referenced in the review.
For this review, we chose to focus specifically on the child-dog bond instead of the adult human-animal bond more broadly, because this is an area that we feel is lacking and in need of further investigation. We include more than two references that relate to the child-dog attachment bond throughout the manuscript.
-Behavioral Perspective… You can change each category into a subsection such as 2.1 physical activity 2.2 learning in the classroom or you can create an overarching them for groups of topics for example “2.1 skills” and then discussing physical activity, learning in the classroom, and learning and development at home
We thank the reviewer for this suggestion. We have made this organizational change in the manuscript.
-Same feedback for Biological Perspective that I had for Behavioral Perspective
We have made these changes in the manuscript.
-Consider changing main subjects to “Behavioral Perspective” and “Biological Perspective”
Done!
-Figure 1: make sure to create an image that can be moved with text and is in line with the rest of the body
We have updated the image formatting so that it can be moved in the text and resized to fit the space (“wrap text” formatting).
-Review requirements for attaching a table and edit Table 1
We have edited Table 1 according to the table requirements.
-Reword conclusion: give brief summary of what was discussed. Change questions into topics/statement of the things that need to be addressed in future
We have re-worded the conclusion section.